# So You Think You Can Scale Up Autonomous Robot Data Collection?

**Suvir Mirchandani, Suneel Belkhale, Joey Hejna,**
**Evelyn Choi, Md Sazzad Islam, Dorsa Sadigh**
Stanford University
https://autonomous-data-collection.github.io

**Abstract:** A long-standing goal in robot learning is to develop methods for robots to acquire new skills autonomously. While reinforcement learning (RL) comes with the promise of enabling autonomous data collection, it remains challenging to scale in the real-world partly due to the significant effort required for environment design and instrumentation, including the need for designing reset functions or accurate success detectors. On the other hand, imitation learning (IL) methods require little to no environment design effort, but instead require significant human supervision in the form of collected demonstrations. To address these shortcomings, recent works in autonomous IL start with an initial seed dataset of human demonstrations that an autonomous policy can bootstrap from. While autonomous IL approaches come with the promise of addressing the challenges of autonomous RL—*environment design challenges*—as well as the challenges of pure IL strategies—*extensive human supervision*—in this work, we posit that such techniques do not deliver on this promise and are still unable to scale up autonomous data collection in the real world. Through a series of targeted real-world experiments, we demonstrate that these approaches, when scaled up to realistic settings, face much of the same scaling challenges as prior attempts in RL in terms of environment design. Further, we perform a rigorous study of various autonomous IL methods across different data scales and 7 simulation and real-world tasks, and demonstrate that while autonomous data collection can modestly improve performance (on the order of 10%), simply collecting more human data often provides significantly more improvement. Our work suggests a negative result: that scaling up autonomous data collection for learning robot policies for real-world tasks is more challenging and impractical than what is suggested in prior work. We hope these insights about the core challenges of scaling up data collection help inform future efforts in autonomous learning.

**Keywords:** autonomous data collection, imitation learning

## 1 Introduction

Enabling robots to acquire skills in the wild from autonomous, self-supervised interaction has been a long-standing goal in robot learning. To this end, a variety of efforts have focused on developing methods for reinforcement learning (RL) in the real-world [1–3]. Despite substantial progress, RL for real-world robotics requires a significant amount of human effort on *environment design*, such as developing reset mechanisms, safe guards, success detectors, and reward functions. These challenges—exacerbated by sample efficiency issues—have constrained the complexity of tasks that are possible with today's methods for real-world RL. As a consequence, many have shifted their attention to imitation learning (IL) methods, which scale much better with task complexity [4, 5]. However, IL methods rely on increasingly large amounts of high-quality human demonstrations as tasks become more diverse and complex, thus shifting the human effort required to *human supervision*—i.e., demands on the time of expert operators. In fact, from "pure autonomous" RL methods to "pure human" IL methods, there exists a spectrum that trades off between environment design effort and human supervision effort. We visualize this spectrum in Fig. 1, where one might expect that by moving from either side towards the middle of this spectrum, the human effort in both environment design and supervision can go down.

8th Conference on Robot Learning (CoRL 2024), Munich, Germany.

A variety of works have attempted to move toward the middle of this spectrum, either by reducing environment design challenges (often increasing supervision requirements) for RL methods [6–10] or decreasing supervision requirements (often increasing environment assumptions) for IL methods [11–19]. Our hope as a field is that somewhere in the middle lies an effort-minimizing approach that will enable robot learning methods to effectively scale.

One proposed middle-ground approach is *autonomous IL*, where we let a policy autonomously collect its own data using a policy trained on an initial amount of human data and iteratively re-train with the successful rollouts [17–19]. Many hope that this approach will finally push us to the bottom of the "U" in Fig. 1 (solid line), since the promise of autonomous IL is to reduce human data collection effort while partially mitigating safety and exploration issues.

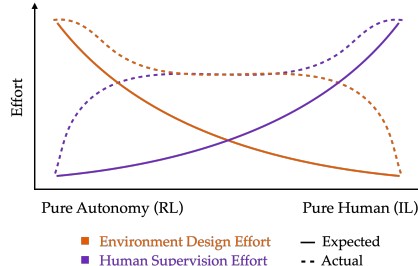

**Figure 1:** Conceptual diagram illustrating the expected vs. actual effort tradeoff for human supervision and environment design.

In this work, we examine the challenges of applying autonomous IL to useful and realistic manipulation tasks in the real world—beyond what is often shown in simpler toy settings. We find that in practice this middle ground approach unfortunately suffers from many of the same scaling challenges as prior work. We observe that the true effort curve looks more like Fig. 1 (dashed line): as we try to reduce environment design effort or reduce human supervision collection effort, we find that the total human effort required to see comparable success rates plateaus.

Our study is organized into two parts. First, in §3, we demonstrate that autonomous IL methods still suffer from high environment design costs—for example, reset functions and success detection. In practice, these costs limit the complexity of tasks that can be tackled with autonomous IL. Second, in §4, we select 7 simulation and real-world tasks where environment design costs can be minimized, and through 10K+ real-world evaluations and over 100 hours of autonomous data collection, we rigorously evaluate a range of autonomous IL methods. While several methods lead to mild performance improvements (on the order of 10%) on top of an IL policy trained on the initial human demonstrations, we consistently find that collecting a few more human demonstrations surprisingly is a more efficient use of total effort. **Our work suggests a negative result: that scaling up autonomous IL for real-world tasks might be much more challenging than what is conceived by the field and what prior work suggest**, given that these methods still require significant environment design effort and underperform simply redirecting this effort to collecting demonstrations. This work sheds light on the true bottlenecks of scaling up data collection, such as finding generalizable solutions to environment challenges and developing methods to scale up human supervision.

## 2 From RL to IL: Preliminaries and Related Work

Here, we introduce the spectrum of robot learning methods from RL to IL, their assumptions, and prior work.

### 2.1 Reinforcement Learning

Reinforcement Learning (RL) methods adopt the model of a standard Markov decision process (MDP) $\mathcal{M} = (\mathcal{S}, \mathcal{A}, \mathcal{T}, R, \rho_0, \gamma)$ consisting of a state space $\mathcal{S}$, continuous action space $\mathcal{A}$, transition function $\mathcal{T} : \mathcal{S} \times \mathcal{A} \times \mathcal{S} \rightarrow [0,1]$, reward function $R : \mathcal{S} \times \mathcal{A} \rightarrow \mathbb{R}$, initial state distribution $\rho_0$, and discount factor $\gamma \in [0,1]$. These methods aim to learn a policy $\pi$ to maximize the expected discounted sum of rewards $J(\pi) = \mathbb{E}[\sum_{t=0}^{\infty} \gamma^t R(s_t, a_t)]$. Implementing RL algorithms in practice is challenging due to a number of factors:

**Success Detector ($\mathcal{S}, R$).** As an evaluation mechanism or for early termination of episodes, it is typical to utilize a success detector $f : \mathcal{S} \rightarrow \{0,1\}$ to detect if a state $s$ falls within a set of terminal states $\mathcal{S}^+ \subset \mathcal{S}$. The success detector $f$ may be learned, scripted, or labeled by a human.

**Reset Mechanism ($\rho_0$).** Generating full episodes assumes the ability to sample from $\rho_0$. This requires access to a reset policy (commonly referred to as a backward policy $\pi_b$) such that following $\pi_b$ leads to a new initial state $s_0' \sim \rho_0$. The existence of $\pi_b$ necessitates that all $s_0' \in \rho_0$ are reachable from any $s \in \mathcal{S}$ at which the episodes terminate. However, this assumption does not hold in the case of irreversible actions. In

practice, it is common to instrument environments with hand-crafted primitives, physical reset mechanisms, or require humans to perform resets [20, 21].

**Stationary Dynamics** ($\mathcal{T}$). In most works, $\mathcal{T}$ is assumed to be stationary. This can significantly restrict the real-world environments on which RL can be used.

Crafting environments that meet the above requirements can be surprisingly challenging in practice. As a result, many works have focused on highly controlled "toy" tasks which—while presenting challenges from a learning perspective—have been selected such that it is easy to learn or specify a success detector and easy to perform resets, or require humans to do resetting and success detection [1, 2, 10, 20, 21]. Thus, these tasks tend to involve relatively simple manipulations (e.g., lifting or pushing objects, opening drawers, etc.) that do not capture the complex manipulations that we eventually hope for robots to accomplish.

## 2.2 Imitation Learning

In contrast, Imitation Learning (IL) methods aim to learn a policy $\pi$ that imitates the behaviors from a dataset $\mathcal{D}$ where each demonstration $\xi \in \mathcal{D}$ consists of state-action transitions $\{(s_0,a_0),...,(s_T,a_T)\}$ [22]. Rather than assuming access to reward $R$, IL generally assumes that each demonstration $\xi$ is given by an expert, and thus attains maximum reward. While avoiding the requirements of a reward function, an automated reset function, and success detector, IL imposes additional key assumptions:

**Data Collection Time.** The creation of $\mathcal{D}$ requires access to expert operator(s) to collect demonstrations. In practice, it is common for the operator to additionally perform resets between episodes.

**Optimal Demonstrations.** The majority of works assume $\mathcal{D}$ is composed of optimal demonstrations. Producing high-quality demonstrations imposes additional burdens on operators, such as practicing before collecting demonstrations, as well as filtering out low-quality demonstrations during data collection [23, 24].

## 2.3 The Middle Ground: Mixed Autonomy Methods

In an attempt to strike a balance between the environment design challenges of pure RL and human supervision challenges of pure IL, several works have proposed methods that mix human demonstrations and learning from autonomous execution. Closer to the left side of Fig. 1, hybrid RL+IL methods use a prior dataset of human demonstrations to guide RL (providing some amount of exploration guidance and sample efficiency gains) [6, 8–10, 25]. Closer to the right side of Fig. 1 are interactive imitation learning (IIL) methods, which allow a human to intervene on a robot's autonomous execution, and use these interventions as a learning signal [11, 13, 17, 26]. In practice, these works still require significant environment design effort (e.g., to prevent unsafe behaviors or instrument success detectors) and human supervision effort (e.g., for resetting environments, or supervising autonomous execution until an intervention is needed).

More recently, interest has grown in autonomous IL methods which self-bootstrap starting from a policy trained on human demonstrations [17–19]. The hope is that these methods can bypass the need for high environment design effort by removing reward and safety constraints. They could also reduce human supervision by only requiring a fraction of the initial data that pure IL would require for the same tasks. In this work, our aim is to stress-test these ideas with various autonomous IL methods as we scale up task complexity.

We provide a general recipe for autonomous IL in Alg. 1. Given a dataset of human demonstrations $\mathcal{D}_H$, a policy $\pi_0$ is trained using algorithm A (Line 2). For $M$ rounds, a new dataset $\mathcal{D}_A^i$ is generated by collecting $N_A$ rollouts (Line 4) with a filter function $F$. A new policy $\pi_i$ is trained using algorithm B on a mixture of the prior datasets specified with

---

**Algorithm 1:** Autonomous IL Recipe

Given rounds $M$, alg. A, alg. B, filter $F$, mixture fn `mix`
1   $\mathcal{D}_H \leftarrow$ Collect $N_H$ human demonstrations
2   $\pi_0 \leftarrow$ Train alg. A on $\mathcal{D}_H$
3   **for** $i$ in $1...M$ **do**
4     $\mathcal{D}_A^i \leftarrow$ Collect $N_A$ rollouts of $\pi_{i-1}$ under filter $F$
5     $\pi_i \leftarrow$ Train alg. B on `mix`$(\mathcal{D}_H,\mathcal{D}_A^1,...,\mathcal{D}_A^i)$

---

a function `mix` (Line 5). The simplest, naïve instantiation is filtered behavior cloning (BC): where both algorithms A and B are BC, the number of rounds $M = 1$, and the filter function $F$ only accepts successful episodes, so as not to pollute the training data with failures (Line 4). Past works have alluded to this idea that naïvely adding successful autonomous rollouts added to the training data in this way should be helpful to performance [17–19]. We systematically test whether this is the case in the single task setting. We first analyze the challenges of satisfying the environment assumptions of Alg. 1 as we scale task complexity

(§3), and then test whether autonomous IL can meaningfully reduce human supervision effort across a variety of variations and design choices in this recipe (§4).

## 3    Challenges of Scaling Up: Analyzing Environment Design

In this section, we provide a case study on the challenges of running autonomous IL in practice on useful real-world tasks. These challenges correspond to satisfying pre-conditions of most autonomous IL algorithms as in Alg. 1, but for concreteness, we limit the discussion to filtered BC as a simple instantiation of Alg. 1: (1) $\pi_0$ must achieve nontrivial success given $N_H$ demonstrations; (2) the success detector $f$ used to define the filter $F$ must be accurate; (3) the environment dynamics $\mathcal{T}$ must be stationary; (4) the reset mechanism to sample from $\rho_0$ must be robust. This case study illustrates that environment design effort, while often underemphasized, is difficult to reduce as we approach more useful and realistic tasks.

**Useful but Feasible Tasks:** *from organizing laundry to folding socks.* To test if autonomous IL delivers on its promise of addressing challenges of IL and RL, we need to select *useful real-world tasks*, where IL and RL techniques struggle. Consider the task of folding laundry: this task requires manipulation of deformable clothing of many shapes and sizes with a broad distribution of initial states (e.g., object configurations). For autonomous IL, this causes a few issues: (1) The initial autonomous policy must achieve nontrivial success rates in order for us to collect *any* autonomous data: thus a large amount of initial human supervision effort (demonstrations) is already necessary for these realistic tasks. Recent works have shown one can achieve challenging tasks similar to laundry folding—albeit with limited generalization, i.e., limited variations in scenes or initial configurations—via imitation learning on thousands of demonstrations [27]. (2) We need to be able to reset the environment to initial configurations that autonomous IL can bootstrap from, which is often infeasible for realistic tasks like laundry folding which have a broad set of initial configurations. (3) Performing controlled evaluations on different models is challenging when the set of initial conditions is so broad. Thus, to even begin to study the problem of autonomous IL, we scope the task down significantly to a more controlled setting: *sock folding* from an arbitrary configuration (see Fig. 2). This task nevertheless represents a step up in difficulty from toy tasks (e.g., folding a square cloth that always begins unfolded). A diffusion-based imitation policy [5] trained on our sock folding task attains only ~30% success trained on 250 human demonstrations.

**Reliable Success Detection:** *from folding socks to hanging mitts.* To train on only successful autonomous data without human supervision, we need a precise and reliable success detector. Without a good success detector, datasets can be polluted with false positives, and controlled evaluation becomes very challenging. Even when limiting ourselves to the sock folding task, simple object shape and area heuristics (which have been used in prior work on toy cloth folding tasks) prove to be insufficient for crisply detecting whether a sock has been folded in half; indeed, autonomous execution inevitably encounters "edge cases" which are challenging even to annotate by hand. We also attempt to train a success classifier on terminal states from a combination of 200 human demonstrations and 700 hand-annotated rollouts of the autonomous policy with an approach similar to [18], still obtaining a validation error of 10-20%, representing the challenge of training success detectors for realistic tasks, even under generous data assumptions, without the use of more specialized or domain-specific techniques. Please see Appx. A for more details.

Despite our attempts to reduce environment design effort, we find it necessary to "shape" the task to fit our requirements—e.g., changing the choice of sock (to be shorter and stiffer) as well as the material of the table to be a higher friction surface (often used in prior work with cloth manipulation [28]). Even with this task

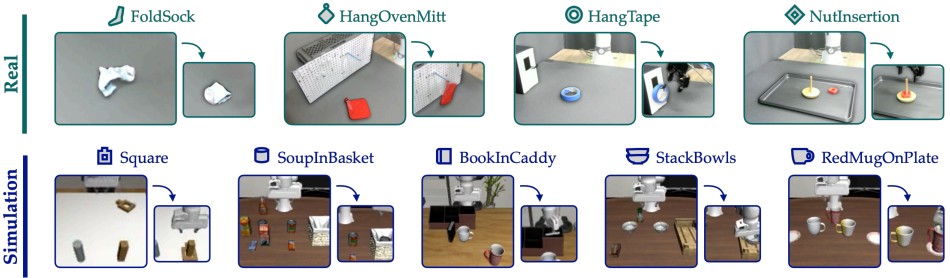

**Figure 2:** Initial and final success states in 4 real-world environments (top) and 5 simulation environments (bottom).

structure, autonomous rollouts yet again produce many edge cases where detecting success is challenging without full state information. For example, a common failure case is a Z-fold of the sock—where the sock would be folded twice in a Z-shape—which is difficult to perceive without depth information. Given these difficulties, we find it necessary to select a task with a more reliable success detector. To maintain task usefulness, we consider hanging a deformable oven mitt with a small loop on a hook (Fig. 2).

**Environment Stationarity:** *from hanging mitts to rigid-body tasks*. While success detection is straightforward on the oven mitt task (and can be done simply by using color thresholds—see Appx. A), we encounter a new, significant challenge in scaling up autonomous data collection for this task: non-stationarity of the environment over time. Fig. 3 shows the success rate of a diffusion policy trained on 200 demonstrations of HangOvenMitt. Note that the success rate of this fixed policy *significantly decreases* over the course of several hundred rollouts in a day. Interestingly, the initial success rates are still reproducible after the object is at rest for several hours. We hypothesize that with continuous interactions and resets, the state of the object becomes marginally more

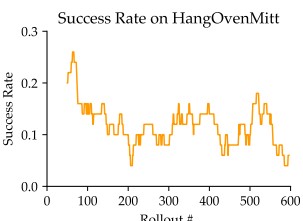

**Figure 3:** Success rate on Hang-OvenMitt over 600 rollouts, with window size of 50 rollouts.

deformed over time, in a way that is restored after the object remained at rest. This confounding factor impedes controlled comparisons between policies. Deformability is just one source of environment non-stationarity: lighting, object textures, and even camera positions can change unpredictably or over time. We need to control for these sources of variation, especially in autonomous IL, where the changes in data distribution can have different implications on the policy. To avoid issues of non-stationarity due to deformability, we find it necessary to move to rigid-body tasks.

**Robust Reset Functions:** *simple rigid-body tasks*. To autonomously collect data, designing a robust reset function is critical for all the tasks we have discussed so far: if the reset fails even 1% of the time, significant human supervision effort is required to monitor the scene. This places several limitations on the tasks we can do with autonomous IL: (1) The reset function must be easier than the task itself. Thankfully, this is true in many complex manipulation tasks, including many tasks with deformable objects. For example, in the case of sock folding, the scene can be reset simply by flinging the sock (from any configuration) whereas the forward task of folding requires much more precision. (2) The policy cannot encounter irreversible states. For the oven mitt reset procedure, we encounter a variety of irreversible states: the mitt gets lodged under the hook or falls out of the robot workspace, at a rate of approximately once every 100 rollouts. This is solved by adding reset instrumentation (specifically, a string attached to the object) such that the robot can pull the object back into a reachable position. This instrumentation vastly increases the robustness of the reset policy at the cost of environment design time. We provide more information on the implementation of the reset functions in Appx. A.

**Summary.** To perform useful tasks in the real world, autonomous IL methods encounter a variety of challenges that consistently keep environment design costs high. Tasks cannot be *too complex* or evaluation and data collection become costly. Success detection and reset functions must also be extremely robust. Environments must be stationary during the course of learning and evaluation. All of these requirements makes it difficult to apply autonomous IL for useful and realistic tasks in the real world.

> *Takeaway*: Scaling task complexity for methods that involve *truly autonomous* real-world data collection remains challenging. Given these environment design challenges, autonomous IL methods are realistically limited to rigid-body tasks with easy and safe reset functions like pick-place, articulated object manipulation, or insertion.

## 4  Challenges of Scaling Up: Analyzing Human Supervision

In §3, we describe environment design challenges that hinder scaling up IL on useful, realistic tasks. Now let us assume that environment challenges are surmountable (i.e., by limiting tasks to simple rigid-body manipulation like pick-place or insertion). In such settings, can autonomous IL methods meaningfully reduce the amount of *human supervision* needed to learn an effective policy? Once again, our attempts to reduce this source of effort lead us to several key challenges in scaling autonomous data collection.

## 4.1 Experiment Overview

We study a variety of instantiations of Alg. 1, from straightforward techniques such as filtered BC (simply rolling out a policy trained on human data and adding successful rollouts back into the training set), to more complex ones such as active learning and offline RL. For the majority of experiments, we use Diffusion Policy [5] as the choice for algorithms A and B given its expressivity and ability to capture multimodal action distributions [5]. We also set $M = 1$ unless otherwise specified, and train models from scratch at each iteration.

- In §4.2, we study the impact of *data scales* (of $N_H$ and $N_A$); and *number of rounds* (setting $M > 1$).
- In §4.3, we investigate *novelty-based reweighting strategies* (more sophisticated mix functions).
- In §4.4, we study *active learning guided by failures* (where $F$ excludes rollouts whose initial states are near previous successes).
- In Appx. B, we provide ablations on *data weights* (upweighting human or autonomous data with mix), *training methods* (modifying B to train from scratch versus fine-tune $\pi_{i-1}$), *policy class* (replacing A and B with ACT [4]), *offline RL* algorithms (where $F$ allows both successes and failures, and B is an offline RL algorithm), and training with *out-of-distribution* autonomous successes (modifying Line 4).

**Task Selection.** Due to the environment challenges described in §3, we limit our analysis to two rigid-body real-world tasks shown in Fig. 2: HangTape (on a hook) and NutInsertion (on a peg) as well as four simulation tasks from LIBERO [29]: SoupInBasket, BookInCaddy, StackBowls, and RedMugOnPlate; and one task from Robomimic [24]: Square. These simulation tasks (shown in Fig. 2) enable a thorough evaluation of design choices for scaling up autonomous data collection while avoiding environment design challenges. Appx. A contains more details on task implementation, success detection, and reset procedures. For each policy evaluation in this work, we perform *100 trials* for real-world tasks and *200 trials* for simulation tasks.

**Data Scale Definitions.** Throughout this section, we abbreviate data quantities as follows: ↓=low, ◇=medium, ↑=high amounts of data for human demonstrations (H) compared to autonomous data (A). For example, ↓H means $N_H$ is a low amount of demonstrations, and ↓H + ↑A means low amount of demonstrations combined with high amount of autonomous data, generated by rolling out a policy trained on ↓H until the requisite number of autonomous successes is reached. Generally, the exact data amounts are not equal between A and H, and we chose the H data scales so that BC performance was roughly in the 20-50% range for ↓ and 50-70% for ↑. Exact data quantities for each environment are provided in plots and in Appx. A.

## 4.2 Diminishing Returns of Filtered BC

In this section, we instantiate filtered BC (naïve autonomous IL), where autonomous rollouts from a policy trained on human data are simply added to the training data. We study the impact of data weights, data scales, and number of collection rounds. In all experiments, we train a Diffusion Policy from scratch on the human-autonomous data mixture. Guided by our results on data weights (Appx. B.2), we train from scratch with 50-50 human-autonomous mixtures for the experiments in this section. Please see Appx. B for additional training details and hyperparameters, and for ablations on training methods (e.g., fine-tuning).

**Human and Autonomous Data Scales**. We now study how the scale of initial human data and the ratio of human data to autonomous data affect performance. In Fig. 4, we compare the low and medium human data regimes across all simulation and real environments, for various ratios of human to autonomous data. Overall, adding autonomous data can modestly improve policy performance on both low and high data regimes (on average 10-20%); but occasionally, it has a minor negative effect (Square, BookInCaddy (↓H), StackBowls (↑H)). *In most cases, any positive effect saturates as autonomous data scales.* In line with prior work in data quality, we suspect that some amount of autonomous data is good for capturing more state diversity, but sometimes that data can also cause action consistency to decrease [23, 30]. *Moreover, adding more human data (purple vs. orange) tends to have a stronger effect than adding autonomous data.*

**Multiple Collection Rounds.** Observing the somewhat positive effect for incorporating autonomous data, one might expect this trend to continue for multiple rounds of autonomous data collection followed by training on the newest round of autonomous data (i.e., when $M > 1$). This procedure is similar to running Reward-Weighted Regression seeded by demonstrations [31]. In Fig. 5, we run autonomous collection for 2–4 rounds in several simulation and real environments. Interestingly, multiple rounds can in fact improve performance (most LIBERO tasks), but just like a single round of collection, it can also saturate performance (BookInCaddy, SoupInBasket (↑H), and real tasks) or even hurt performance (Square). We suspect that

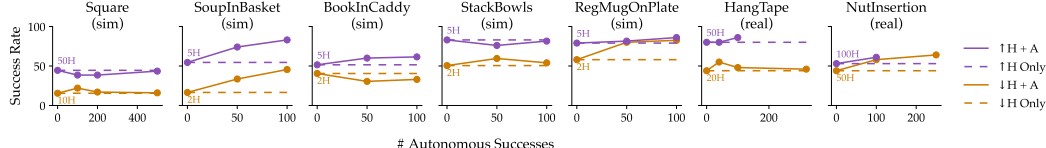

**Figure 4:** Performance of filtered BC over difference scales of human and autonomous data with 50-50 co-training on various simulation and real environments. More autonomous data often helps (left to right within each line plot), but having more initial human data (labeled H) generally has a stronger effect (purple vs. orange).

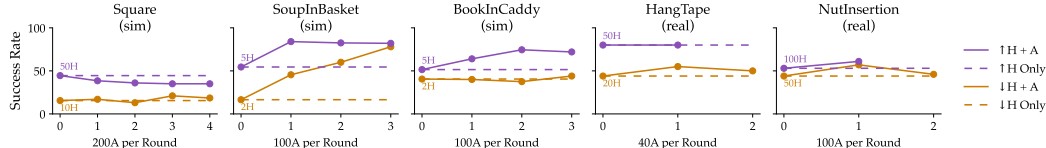

**Figure 5:** Multiple Rounds (in order, left to right) of filtered BC in simulation and real environments. We see either saturating increases or decreases in performance.

unlike LIBERO tasks, in Square there are challenging bottleneck states in which subtle variations in the action distributions in autonomous data can greatly affect policy performance (see Appx. B). Furthermore, with the exception of SoupInBasket (↓H), *multiple rounds seem to plateau in performance after the initial improvement, and collecting a few additional human demonstrations often beats this plateaued success rate*. Thus, multiple rounds of autonomous collection might not be worth the added effort.

### 4.3 Inconsistent Response to Novelty-Based Reweighting

Adding successful autonomous data to human data can yield modest improvements in some cases. Instead of simply adding *all* successful autonomous data to the training dataset, one might ask the question, is all the autonomous data equally *useful*? Specifically, what states and actions are valuable to learn from? Building on prior work [13, 23, 30], we consider using *state novelty* as a metric for the utility of a new state-action pair. Our intuition is that more common states are either redundant or potentially have conflicting actions; this data is likely less useful than more novel states. Building on prior work, we inform the sampling weights of the autonomous dataset in the mixture function `mix` based on different notions of novelty. We use two measures of novelty in Fig. 6 on the Square task; see Appx. B.5 for formal definitions.

1. *Action Novelty*: Measure novelty as proportional to the variance in action predicted by an ensemble of policies trained on the same data.
2. *Image Embedding Novelty*: Measure novelty as proportional to the variance in image embeddings from an ensemble of vision encoders of policies trained on the same data.

**Figure 6:** Policy performance in simulation (Square) across different novelty-based reweighting schemes, using both action-variance and image embedding-variance with ensembles of size 3 as novelty metrics.

Both novelty metrics measure policy uncertainty, but in action novelty the uncertainty is more action-driven, whereas embedding novelty is more state-driven. For both action- and embedding-based novelty metrics, reweighting has a slight positive effect on performance compared to the naive strategies for ↓H, a more notable effect for ◇H, and a slight decrease for ↑H. Thus, *novelty-reweighting provides inconclusive results and still underperforms just adding a bit more human data*. Yet, these results confirm our intuition that in many cases, much of the autonomous data is redundant and can be filtered out without decreasing performance.

### 4.4 Inability to Learn from Failure Data

Reweighting the autonomous dataset (e.g., using novelty) still depends on the distribution of states visited by the autonomous policy. And in the novelty-weighting case study, by setting *F* such that only *successful* autonomous rollouts are included in the autonomous dataset, we are critically biasing this distribution toward states where the policy is already proficient. Does learning from *failure data* yield any improvement? We study two approaches to learn from failures: active learning and offline RL (Appx. B).

**Active Learning Guided by Failures.** Are there more intelligent ways to target the new data we collect? One such approach, inspired by active learning methods, is to use autonomous failure data to generate a distribution of *failure* states to intelligently query a user. Querying a user for a single expert action at arbitrary failure states is challenging in practice, and it is often easier for a user to provide a complete demonstration. Therefore, we implement a more practical approach: sampling from the distribution of *initial states* from failed autonomous rollouts to query for complete demonstrations. In Fig. 7, we compare (1) using the autonomous policy to collect new demonstrations near the failure initial states (Near-Failure A), and (2) querying a human for demonstrations initialized at these states (Near-Failure H).

Incorporating near-failure autonomous data has no positive effect beyond the naïve method of random selection. However, collecting new human demonstrations near autonomous failures (Near-Failure H) does yield improvement over an equal number of randomly selected human demonstrations (Random). Similar to the filtered BC case study, *collecting more human demonstrations tends to yield larger improvement than even high amounts of autonomous data, especially if those demonstrations are collected from autonomous failures' initial states.*

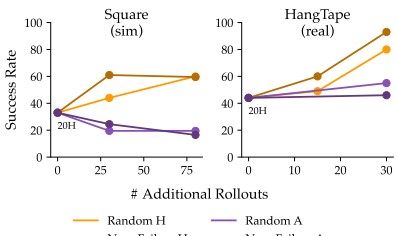

**Figure 7:** Results for Square and HangTape, querying either an autonomous policy (Near-Failure A) or human (Near-Failure H) for new demonstrations near the set of autonomous failures' initial states. Adding targeted human data helps much more than equivalent amounts of random human data, targeted autonomous data, or random autonomous data.

**Summary.** We find that the most salient factors leading to policy improvement for autonomous IL are, in order: (1) the amount and utility of new human demonstration data and (2) the availability of some amount of autonomous data. Data weights, amount and rounds of autonomous collection, and novelty-based reweighting strategies all have little effect in comparison. Repeatedly, we find that additional human data is significantly better than access to even ten times the amount of autonomous data. This suggests that even under the assumption of no environment challenges, many autonomous IL methods struggle to match the performance of simply incorporating a small amount of additional human data.

> *Takeaway*: Even for simple tasks that do not pose environment challenges, both straightforward and more advanced autonomous IL methods only modestly improve performance, and often plateau as autonomous data increases.

## 5    Discussion

In this work, we take a practical look at the challenges involved in scaling autonomous IL to complex tasks. While autonomous IL does not require all of the assumptions of fully autonomous methods like RL, we affirm that they still require immense environment design effort which scales with task complexity. Designing robust resets and precise success detectors remains an open challenge for such complex tasks, as does dealing with environment non-stationarity. These factors are critical to conducting meaningful evaluations and deploying algorithms with any level of autonomy in the real world, and so scaling robot learning requires advances in how we can mitigate or amortize environment design effort. For example, a direction for improvement is in learning general purpose success detectors using foundation models. Even assuming we can reduce environment design effort, in this work our best attempts at learning from the autonomous data could not match the improvement of marginally increasing human supervision effort with traditional IL. We hope future work will build on the insights in our work to extract as much as possible from autonomous data collection—for example, by guiding both human and autonomous data collection based on insights (e.g., novelty, failure states) gleaned from the autonomous trials.

**Limitations**. While our study of autonomous IL considers a wide variety of methods, we primarily consider single-task imitation learning, and the performance of autonomous IL methods may differ in multi-task settings. Multi-task environments could also enable learning the reset and the task at the same time, alleviating some environment design effort. Finally, we focus on settings where models are trained from scratch; future work should study effects of large-scale pre-training in the autonomous IL setting.

**Acknowledgments**

Toyota Research Institute provided funds to support this work. We are also grateful for additional support from NSF Award 1941722 and 2218760, DARPA Award W911NF2210214, ONR Award N00014-22-1-2293, and the Stanford Human-Centered AI Institute Hoffman-Yee Grant. Finally, we thank Jensen Gao, Jennifer Grannen, Hengyuan Hu, Siddharth Karamcheti, Priya Sundaresan, and other Stanford ILIAD lab members for useful discussions and feedback.

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

# A    Task Details

In this section, we give additional information on the tasks studied in this work. We give verbal descriptions in Appx. A.1, definitions of data scales in Appx. A.2, and details on the evaluation procedures in Appx. A.3.

## A.1    Task Descriptions

- *FoldSock*. Fold a sock (with random configuration) neatly in half.
- *HangOvenMitt*. Hang an oven mitt (with random position and orientation) on a hook (fixed position).
- *HangTape*. Hang a roll of masking tape (with random initial position) on a hook (fixed position).
- *NutInsertion*. Insert a plastic nut (with random initial position) on a peg (fixed position).
- *Square*: Insert a square nut on a square peg (from [24]).
- *SoupInBasket*: Place a small soup can into a basket (from [29]).
- *BookInCaddy*: Place a book into a narrow book caddy (from [29]).
- *StackBowls*: Stack two bowls together and place both on a plate (from [29]).
- *RedMugOnPlate*: Put a red mug on a specific plate (from [29]).

We include an illustration of initial state distributions, sample initial and successful states, and sample camera observations for the NutInsertion and HangTape tasks in Fig. 8.

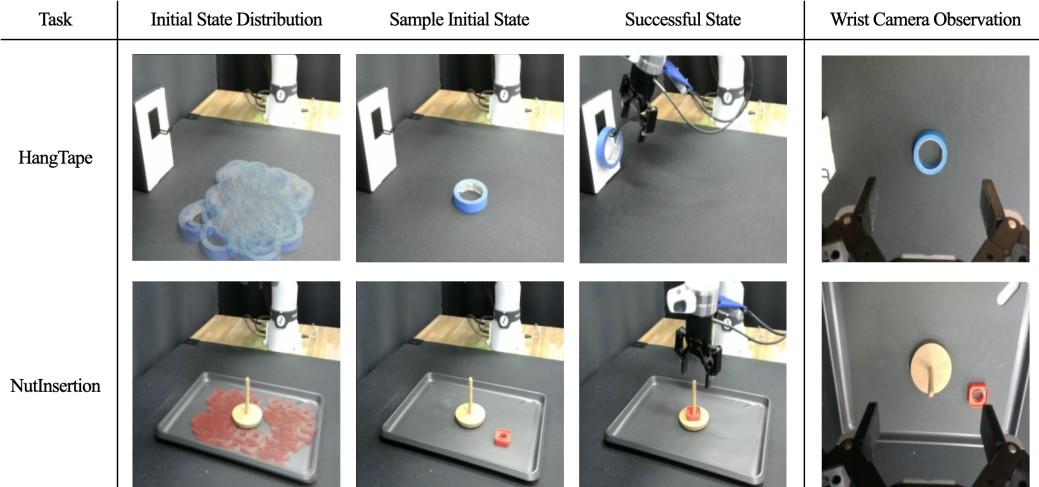

**Figure 8:** For the HangTape and NutInsertion tasks, we include scene images depicting the initial state distribution (using an overlay of initial state samples), a sample initial state, a successful state, and a view of the initial state from the wrist camera's perspective.

## A.2    Data Scale Definitions

For concision, and to focus on trends, we abbreviate data scales (i.e., number of demonstrations) as low ($\downarrow$), medium ($\diamond$), and high ($\uparrow$) for each of human demonstrations (H) and autonomous rollouts (A). Due to the fact that tasks vary widely in difficulty, the absolute value of demonstrations for each data scale varies per task. We include these values in Table 1.

**Example.** To generate the training set for the $\downarrow$H + $\downarrow$A setting on the NutInsertion task, we do the following:

- Collect 50 human demonstrations from randomly sampled initial states.
- Train an initial policy on the human demonstrations to convergence (approximately 47% success rate).
- Collect 100 successful autonomous rollouts (by rolling out the policy over 200 times and filtering out the failures).

## A.3    Evaluation Procedure

Unless otherwise specified, all success rates in this work are calculated by uniformly sampling an initial state $s_0 \sim \rho_0$ and rolling out the learned policy under consideration until either a success state is achieved or

| Env | ↓H | ◇H | ↓A | ◇A | ↑A |
|---|---|---|---|---|---|
| FoldSock | 100 | 250 | — | — | — |
| HangOvenMitt | 200 | 500 | — | — | — |
| HangTape | 20 | 50 | 40 | 100 | 320 |
| NutInsertion | 50 | 100 | 100 | 250 | —— |
| Square | 10 | 50 | 100 | 200 | 500 |
| SoupInBasket | 2 | 5 | 50 | — | 100 |
| BookInCaddy | 2 | 5 | 50 | — | 100 |
| StackBowls | 2 | 5 | 50 | — | 100 |
| RedMugOnPlate | 2 | 5 | 50 | — | 100 |

**Table 1:** Legend of data scales for each environment.

a maximum time horizon is reached. For all simulation results, we perform 200 trials. For all real results, we perform 100 trials.

## A.4 Success Detection and Resets

In this section, we provide additional details and rationales for the success detection and reset pipelines that we used in our real-world tasks. For tasks in simulation, success detection and resets were provided by the environment.

### A.4.1 Success Detection

- *FoldSock*. As we found that scripting a sock-foldedness detector based on heuristics like object shape and area produced false positive and false negative rates on the order of 20%, we attempted to train a success classifier using a similar procedure to [18]. We assemble a training dataset of 200 human demonstrations (which are curated to be always successful) and roughly 700 rollouts from the autonomous collection policy (which we hand-label as success or failure). The training set includes 301 successful trajectories and 438 failure trajectories, and we sample from the end from each rollout (last 5 images) to yield images to associate with the success/failure label. We train a ResNet-18-based architecture with a binary classification head. The validation error of the trained classifier is approximately 15%.
- *HangOvenMitt*, *HangTape*, *NutInsertion*. These tasks include bottlenecks which must be reached in order to succeed at the task: hanging an object on a hook or placing an object on a peg. Therefore, successes and failures are easy to separate. For simplicity, we use scripted rules similar to prior work (e.g., [10]). Specifically, we use color thresholds at pixels located at these bottlenecks, coupled with the condition that the gripper must be open for five steps prior to success. This ensures that the agent has placed the relevant object at the bottleneck in question. We manually verify that the error rate of this detection scheme is near-zero. While we could in principle train classifiers to learn the boundary between success and failure, our higher-level message is that environment challenges like success detection can be a bottleneck for realistic tasks like *FoldSock*, and can influence task design to make tasks more constrained such that success and failure are easy to detect. In §4, we set aside environment challenges (i.e., assume that robust success detection is available) in order to study whether autonomous IL can reduce human supervision challenges.

### A.4.2 Resets

In our study, we use object-centric primitives of various complexity to perform resets. Instrumenting environments with hand-crafted primitives, physical reset mechanisms, or requiring humans to perform resets is a common technique in real-world reinforcement learning [20, 21]. As we illustrate in §3, the human effort of environment design (e.g., by instrumenting the environment to make reset primitives possible) remains when we utilize autonomous IL methods, and these can get more involved as we move towards more useful and complex tasks.

- *FoldSock*. We reset the scene by flinging the sock: locating the sock using a segmentation pipeline (GroundingDINO [32] + FastSAM [33]), picking it up using a top-down grasp, bringing it to the center of the workspace, and executing a fling primitive to randomize its configuration for the next episode.

- *HangOvenMitt*. The final state of the mitt has two cases—in the case of success, the mitt is hanging and the mitt can be pulled off the hook by replaying a pre-recorded trajectory; in the case of failure, the mitt is pulled back to a reachable location via a string attached to the robot—and in both cases, the mitt's location is then randomized using a parameterized pick-and-place primitive.
- *HangTape*. We follow a similar procedure as in HangOvenMitt: if the tape is on the hook (i.e., the previous episode was successful), we replay a pre-recorded trajectory to pull it off of the hook. Otherwise, we detect the location of the tape using a simple color mask and execute a pick-and-place primitive to randomize its initial location for the next episode.
- *NutInsertion*. We once again utilize the fact that the final state of the previous episode is either a success, for which the nut can be removed from the peg using a pre-recorded trajectory, or a failure, for which the nut's location can be randomized using a pick-and-place primitive.

## B    Analyzing Human Supervision: Additional Results

In this section, we provide further details on the results in §4 of the main text. In Appx. B.1, we ablate the choice of training from scratch on human-autonomous mixtures (the recipe used in all experiments in the main text). We also provide additional details regarding training with different data weights (Appx. B.2), data scales (Appx. B.3), policy class (Appx. B.3.1), number of rounds (Appx. B.4), and novelty-based reweighting (Appx. B.5), active learning from failures (Appx. B.6), and offline RL (Appx. B.7). While experiments in the main text focus on autonomous data collected in-distribution, we provide additional experiments in Appx. B.8 on training with autonomous data collected from out-of-distribution (OOD) scenarios. Finally, we provide qualitative examples of human and autonomous trajectory distributions in Appx. B.9.

### B.1    Training from Scratch vs. Fine-tuning

All of the models trained on human-autonomous data mixtures in §4 are trained from scratch until convergence. In this subsection, we justify this choice by comparing training from scratch to methods involving fine-tuning.

Specifically, we focus on a single round of autonomous collection for the Square task in simulation. Unless otherwise specified, each model is trained on a mixture of 50% autonomous, 50% human data. We compare the following training recipes:

- *Scratch*: Train a new model from scratch on the human-autonomous mixture.
- *Fine-tune*: Fine-tune the autonomous policy checkpoint that generated the autonomous data on the human-autonomous mixture.
- *Pre-train Autonomous + Fine-tune*: Pre-train a policy from scratch on the autonomous data only, and then fine-tune on the human-autonomous mixture.
- *Scratch Add*: Directly aggregate human and auto data in one dataset (no explicit 50-50 sampling), and train from scratch on this dataset.

In Table 2, we find that training from scratch, fine-tuning from the base policy, and training on combined human and auto datasets all perform comparably. In fact, training methods seem to matter much less than the amount of autonomous data provided. Therefore, for simplicity, we use the *Scratch* training method for all other experiments in the main text.

| Method | $\diamond$H + $\downarrow$A | $\diamond$H + $\diamond$A | $\diamond$H + $\uparrow$A |
|---|---|---|---|
| Scratch | 69% | 61.5% | 79.5% |
| Fine-tune | 68.5% | 66% | 67.5% |
| Pre-train Auto + Fine-tune | 68.5% | 69.5% | 73.5% |
| Scratch Add | 68.5% | 66% | 77.5% |

**Table 2:** Comparing different training methods on Square in simulation, for medium amounts of human data ($\diamond$H) but for increasing amounts of autonomous data ($\downarrow$A to $\diamond$A to $\uparrow$A). All methods perform equivalently in each data regime.

## B.2 Human and Autonomous Data Weights

Our experiments on Data Weights study the impact of relative sampling weights of human-to-autonomous data in the training mixture (i.e., changing `mix`). These experiments keep the amount of autonomous data fixed ($\downarrow$A) and investigate if success rate changes for two scales of human data ($\downarrow$H and $\diamond$H) at different sampling ratios (75-25, 50-50, 25-75). We include these results in table form in Table 3 and Table 4. We find that changing the training data weights has almost no impact for a given data scale. This is line with expectations from prior work when using importance weighted objectives with highly expressive models [34]. Guided by these results, we use the simple training from scratch setting with 50-50 human-autonomous mixtures for the remaining experiments in §4.

| Env | $\downarrow$H 75-25 | $\downarrow$H 50-50 | $\downarrow$H 25-75 | $\diamond$H 75-25 | $\diamond$H 50-50 | $\diamond$H 25-75 |
|---|---|---|---|---|---|---|
| Square | 15.5% | 22% | 21% | 37.5% | 38.5% | 41% |
| SoupInBasket | 37% | 33.5% | 40.5% | 72% | 74% | 77.5% |
| BookInCaddy | 28.5% | 30.5% | 36.5% | 58% | 60% | 62% |
| StackBowls | 53% | 59.5% | 57% | 69.5% | 76% | 68.5% |
| RedMugOnPlate | 80% | 80% | 83% | 82.5% | 81.5% | 82% |

**Table 3:** Different training weightings of human to autonomous data in simulation have negligible effects.

| Env | $\downarrow$H 75-25 | $\downarrow$H 50-50 | $\downarrow$H 25-75 |
|---|---|---|---|
| HangTape | 47% | 55% | 57% |
| NutInsertion | 59% | 58% | 48% |

**Table 4:** Different training weightings of human to autonomous data in real have negligible effects.

## B.3 Human and Autonomous Data Scales

Our experiments on Data Scales (Fig. 4) use a 50-50 mixture and examine how success rate is impacted by the scale of initial human data and the ratio of human to autonomous data. We include the results in table form in Table 5. Including some amount of autonomous data tends to have mild positive effects in most cases, though these effects generally saturate as autonomous data scales. Increasing the scale of human data generally has a stronger effect than adding autonomous data.

| Env | $\downarrow$H | $\downarrow$H + $\downarrow$A | $\downarrow$H + $\uparrow$A | $\diamond$H | $\diamond$H + $\downarrow$A | $\diamond$H + $\uparrow$A |
|---|---|---|---|---|---|---|
| Square | 15.5% | 22% | 16% | 44.5% | 38.5% | 43.5% |
| SoupInBasket | 16.5% | 33.5% | 45.5% | 54.5% | 74% | 83% |
| BookInCaddy | 40.5% | 30.5% | 33% | 51.5% | 60% | 61.5% |
| StackBowls | 50.5% | 59.5% | 54% | 83% | 76% | 81.5% |
| RedMugOnPlate | 58% | 80% | 82.5% | 79% | 81.5% | 86% |
| HangTape | 44% | 55% | 46% | 80% | 80% | 86% |
| NutInsertion | 44% | 58% | 64% | 53% | 61% | — |

**Table 5:** Scales of human data compared to autonomous data for 50-50 co-training on various simulation (top) and real (bottom) environments. More autonomous data often helps, but having more human data generally has a stronger effect.

### B.3.1 Human and Autonomous Data Scales under Different Policy Classes

In this section, we provide additional results on Data Scales using a 50-50 mixture, keeping the task the same but testing two different policy classes: Diffusion Policy (DP) [5] and Action Chunking with Transformers (ACT) [4]. Both methods are capable of modeling diverse action distribution modes. While ACT underperforms DP in this task, the effects on success rate when re-training with different scales of autonomous data are largely similar: there is mild improvement which appears to plateau. The compatible results on ACT and Diffusion Policy suggest that our observations are not unique to the policy class.

| Env | Method | ↓H | ↓H + ↓A | ↓H + ↑A | ◇H | ◇H + ↓A | ◇H + ↑A |
|---|---|---|---|---|---|---|---|
| HangTape | DP | 44% | 55% | 48% | 80% | 80% | 86% |
| HangTape | ACT | 26% | 32% | 27% | 32% | 44% | 40% |

**Table 6:** Scales of human data to autonomous data for 50-50 co-training on the HangTape environment when varying the policy class between Diffusion Policy (DP) [5] and Action Chunking with Transformers (ACT) [4]. Similar trends exist between the two policy classes: autonomous data often helps, but no more than additional human data, and the improvement quickly plateaus.

We choose Diffusion Policy for the remainder of experiments in this work because it is a state-of-the-art IL method and has the same policy class as a state-of-the-art offline RL method, IDQL, which we look at in §4.4.

## B.4 Multiple Collection Rounds

Our experiments on Multiple Collection Rounds (Fig. 5) measure if any positive effects of autonomous data continue over multiple iterations. Specifically, we replace the autonomous data in the training mixture with the latest round of autonomous data collection, and re-train the model from scratch. The amount of autonomous data is kept constant at each round (◇A; ↑A for LIBERO tasks). We investigate the effects of multiple collection rounds at multiple scales of human data (↓H and ◇H) in simulation and at the ↓H scale in real. We present the results in table form in Table 7 and Table 9, generally observing plateaus in performance after an initial improvement in the first iteration. Interestingly, in the Square task, we observe a slight *decrease* in performance. Unlike the LIBERO tasks, Square contains a more challenging bottleneck state, and we hypothesize that subtle variations in the action distributions over multiple rounds of autonomous data collection and training may amplify this challenge. As evidence, in Table 8, we examine the "staged" success rate in Square over multiple iterations: note that the subtask for "moving the square" increases in success rate while the full task (which includes the insertion bottleneck) decreases in success rate.

| Env | Base | Round 1 (◇A) | Round 2 (◇A) | Round 3 (◇A) | Round 4 (◇A) |
|---|---|---|---|---|---|
| Square (↓H) | 15.5% | 17% | 13% | 21% | 18.5 |
| Square (◇H) | 44.5% | 38.5% | 36% | 35% | 35% |
| SoupInBasket (↓H) | 16.5% | 45.5% | 60% | 78% | — |
| SoupInBasket (◇H) | 54.5% | 84% | 82.5% | 82% | — |
| BookInCaddy (↓H) | 40.5% | 40% | 37.5% | 44% | — |
| BookInCaddy (◇H) | 51.5% | 64% | 74.3% | 72% | — |

**Table 7:** Multiple Rounds of autonomous collection using medium autonomous data (◇A) and training in simulation (↓H and ◇H). We see either saturating increases or decreases in performance.

| Stage | Base | Round 1 (◇A) | Round 2 (◇A) | Round 3 (◇A) | Round 4 (◇A) |
|---|---|---|---|---|---|
| Moves Square | 67.5% | 99.5% | 100% | 100% | 94.5% |
| Full Success | 44.5% | 38.5% | 36% | 35% | 35% |

**Table 8:** Multiple Rounds of autonomous collection in Square (↓H), illustrating the success rate for an intermediate stage (moving the square) and the full task.

| Env | Base | Round 1 (◇A) | Round 2 (◇A) |
|---|---|---|---|
| HangTape (↓H) | 44% | 55% | 50% |
| NutInsertion (↓H) | 47% | 57% | 46% |

**Table 9:** Multiple Rounds of autonomous collection using medium autonomous data (◇A) in real for HangTape and NutInsertion. We see that even though success rates improve in Round 1, they do not improve in Round 2.

## B.5 Novelty-Based Reweighting Strategies

In §4.3, we consider if state novelty can be used as a proxy to extract more useful autonomous data, and form the basis for a sampling weight. In this section, we provide more details on these novelty measures. Given an ensemble of policies $\mathcal{E} = \{\pi_1, \pi_2, ..., \pi_N\}$, we instantiate two measures of novelty building on ideas from prior work [13, 23, 30].

1. *Action Novelty*: Measure state novelty as proportional to the variance in the mean action predictions. This variance can be measured by an ensemble of policies trained on the same data:

$$\text{ActionNovelty}(s) = \sum_{i=1}^{N_\mathcal{A}} \text{Var}_j(\mu_{ji})$$

where $\mu_j$ is the mean of the predicted action distribution $\pi_j(s)$ and $N_\mathcal{A}$ is the number of action dimensions.

2. *Image Embedding Novelty*: Measure state novelty as proportional to the variance in image embeddings produced by an ensemble of vision encoders (i.e., the encoders from each policy in $\mathcal{E}$):

$$\text{EmbeddingNovelty}(s) = \sum_{i=1}^{N_h} \text{Var}_j(h_{ji})$$

where $h_j = \text{enc}_j(s)$ (i.e., the embedding from the encoder associated with policy $\pi_j$) and $N_h$ is the number of embedding dimensions.

Given a novelty measure, we assign the training weight for state $s$ to be proportional to $\exp(\text{Novelty}(s)/\beta)$ where $\beta$ is a temperature hyperparameter.

## B.6 Active Learning Guided by Failures

In §4.4, we examine if we can target data collection by utilizing initial states of failed autonomous rollouts. We provide performance trends for policies trained on different amounts of additional human and autonomous data, both targeted and random, when added to the initial ↓H dataset (10 demonstrations in the case of Square and 20 demonstrations in the case of HangTape). We see consistently that random and targeted human data collection outperforms the same amount of random and autonomous data, and also has a higher slope. In Square, there appears to be a dropoff in the relative improvement of targeted human data above random human data. Neither random nor targeted autonomous data improve upon the initial policy in Square, and the improvements from autonomous data are mild in HangTape.

## B.7 Offline RL with Autonomous Successes and Failures

One can argue that failure data has more to offer than just initial states as in §4.4. We turn to offline RL to learn *directly* from both successful and failure examples: modifying $F$ to accept both successes and failures, and setting B to be an offline RL algorithm. We use Implicit Diffusion Q-Learning (IDQL) [35], a state of the art offline RL algorithm, that uses both success and failure data to learn a Q-function expectile, and then uses this to sample high Q-value actions from a generative actor. To not introduce even more environment challenges, we use sparse rewards provided by the same success detection function. We use Diffusion Policy as the generative actor, resampling actions under the Q-function at each time step. In Fig. 9, we compare IDQL to DP trained on successful autonomous data (BC) and a mixture of successful and failure data (SUB). We observe that incorporating failures

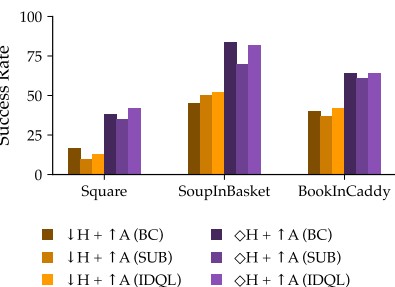

**Figure 9:** Offline RL results, comparing IDQL trained on mixed success and failure data to the naïve autonomous IL strategy (BC), and a suboptimal (SUB) version of naïve autonomous IL trained on both successes and failures. IDQL matches BC and slightly beats SUB.

through IDQL does not outperform naïve autonomous IL, and only slightly outperforms the suboptimal autonomous IL trained on success and failure data. This could be because IDQL struggles to learn a good

Q-function estimate from such a small amount of data and such a high dimensional state space (images). These findings are consistent with prior offline RL results in practice [36].

## B.8 Training on Out-of-Distribution Autonomous Successes

The experiments in the main text focus on training with autonomous data that is collected from *in-distribution* initial states (i.e., initial states are sampled from $\rho_0$ uniformly, or in the case of the active learning experiments, a reweighted version of $\rho_0$). In this section, we examine possible benefits from training on successful autonomous data from out-of-distribution (OOD) scenarios. More specifically, we generate the autonomous data by rolling out the initial policy from a new initial distribution $\rho_0'$ and collect autonomous successes which are the result of the policy generalizing to the new distribution.

In Table 10, we examine the impact on success rates when adding OOD autonomous data in the HangTape task. Specifically, we collect OOD autonomous data where one of two factors is varied compared to the initial distribution: the object (i.e., the tape is changed to a different roll of tape with a different color) and the distribution of initial object positions (i.e., the initial locations are sampled at an expanded outer boundary of the original distribution). When adding 50 successful autonomous rollouts from either of these OOD conditions to 50 in-distribution human demonstrations, we find positive impacts both in-distribution and in the OOD conditions. We see a similar trend in Table 11 on the NutInsertion task, where we collect autonomous data in OOD initial positions (i.e., the initial locations are from an expanded outer boundary) and find that both in-distribution and OOD performance improves.

These insights suggest that OOD autonomous data—i.e., successes that are the result of generalization in the initial policy—may be valuable, at the cost of potentially increasing environment design effort to change the initial state distribution of the environment.

| Data Mixture | Success (ID) | Success (OOD Position) | Success (OOD Object) |
|---|---|---|---|
| 50 H (ID) | 80% | 13% | 27% |
| 50 H (ID) + 50 A (OOD Position) | 90% | 23% | — |
| 50 H (ID) + 50 A (OOD Object) | 83% | — | 51% |

**Table 10:** Success rates both in-distribution (ID) and out-of-distribution (OOD) for policies trained on mixtures of in-distribution human data and OOD autonomous data on the HangTape task.

| Data Mixture | Success (ID) | Success (OOD Object) |
|---|---|---|
| 50 H (ID) | 44% | 40% |
| 50 H (ID) + 50 (OOD Object) | 52% | 50% |

**Table 11:** Success rates both in-distribution (ID) and out-of-distribution (OOD) for policies trained on mixtures of in-distribution human data and OOD autonomous data on the NutInsertion task.

## B.9 Qualitative Examples of Human and Autonomous Data Distributions

In this section, we take a qualitative look at the data distributions of teleoperated human demonstrations compared to autonomous rollouts from policies trained on the human data. We additionally compare the distribution of rollouts from policies co-trained on human and autonomous data.

Fig. 10 illustrates sample initial state distributions from these three categories. In the left column, we superimpose initial states from human teleoperated demonstrations; these initial states are sampled from the initial state distribution of the task. These correspond to data sources for the ↓H settings of HangTape and NutInsertion (20 and 50 demonstrations respectively). In the middle column, we sample initial states from *successful* autonomous rollouts from a Diffusion Policy trained on the human data. These policies are used as the autonomous data collection policies. Note that only a random sample of the successful autonomous data is shown for visualization purposes (a sample of 20 and 50 for HangTape and NutInsertion respectively). Finally, in the right column, we show sample initial states from successful rollouts of a Diffusion Policies co-trained on the human data and autonomous data (with 50-50 data weights). These policies correspond to the ↓H + ↓A settings from §4.2.

| Task | Initial States of Human Demonstrations | Sample Initial States of Successful Autonomous Rollouts | Sample Initial States of Successful Rollouts After 50-50 Co-training |
|------|----------------------------------------|--------------------------------------------------------|----------------------------------------------------------------------|
| HangTape | | | |
| NutInsertion | | | |

**Figure 10:** Comparison of initial states from human demonstrations, autonomous rollouts, and rollouts from policies co-trained on human and autonomous data. The left column shows, from the wrist camera perspective, superimposed initial states from human data. These initial states are sampled from the initial state distribution of the task, and correspond to the data for the ↓H setting. Specifically, this corresponds to 20 demonstrations for HangTape and 50 demonstrations for NutInsertion. In the middle column, we illustrate initial states from sampled successful rollouts of the autonomous collection policy (trained on the human data). In the right column, we illustrate initial states from successful evaluation rollouts from the ↓H + ↓A policy, which is co-trained with a 50-50 mixture of human and autonomous data. Note that, for visualization purposes, the middle and right columns show same number of sampled successful initial states as there are demonstrations in the left column.

In Fig. 11, we similarly illustrate trajectories (end-effector positions) for human demonstrations, sampled successful autonomous rollouts, and sampled successful rollouts of policies trained on human+autonomous data. From Fig. 10, we see a narrowing effect in the distribution of successful initial states, which is more pronounced in the HangTape environment. The policy trained on human demonstrations learns to interpolate between initial locations of the tape that are represented in the human data, especially towards at the center of the distribution. When the policy is re-trained with a mixture of human data and autonomous data, the spread in the distribution of initial states appears to get reduced. However, note that we observe mild overall increases in success rate from autonomous data, and so this is likely due to the policy becoming slightly more robust towards the center of the distribution.

In Fig. 11, we observe an increased homogenization in the successful trajectory paths. This extends beyond just the initial state distributions; note that in both the HangTape and the NutInsertion task, the segments of the trajectories before grasping the object are straighter and less diverse than the corresponding segments in the human data. Additionally, note that the strategies used post-grasp to place the object at its final location (hanging the tape on the hook in the case of HangTape, or placing the nut on the peg in the case of NutInsertion) become more consistent in the autonomous data (as well as the policy co-trained on autonomous data) compared to the human demonstrations.

## C    Training Hyperparameters

For all simulation experiments, we train using Diffusion Policy [5] with the hyperparameters in Table 12 and Table 13. Our real-world experiments use the same hyperparameters, except with an observation history of 1, a step embedding dimension of 128, and 2000 warmup steps. We train policies for the HangTape task for 400K steps and policies for the NutInsertion task for 500K steps. For our ACT experiments, we use the default hyperparameters from [4] except with a chunk size of 16. We execute 8 actions for each inference step at execution time. For the HangTape task, we train policies with 20 human demonstrations for 200K steps and policies with 50 human demonstrations for 400K steps based on model selection between 200K, 400K, and 500K steps.

**Figure 11:** Illustration of trajectories (as 3D end-effector paths) from various policies, with green representing the start of the trajectory and blue representing the end. For reference, we show the scene setup with a sample initial object location in the leftmost column. The second column illustrates human teleoperated demonstration trajectories. The third column illustrates successful autonomous rollouts from a Diffusion Policy trained on the human demonstrations ($\downarrow$H). The fourth column illustrates successful rollouts from a Diffusion Policy co-trained on human data and autonomous data ($\downarrow$H + $\downarrow$A).

| | |
|---|---|
| Diffusion Architecture | Conv1D UNet |
| Prediction Horizon | 16 |
| Observation History | 2 |
| Num Action | 8 |
| Kernel Size | 5 |
| Num Groups | 8 |
| Step Embedding Dim | 256 |
| UNet Down Dims | [256, 512, 1024] |
| Num Diffusion Steps | 100 |
| Num Inference Steps | 10 |
| Inference Scheduler | DDIM |
| Observation Input | FiLM |
| Image Encoder | ResNet-18 |
| Image Embedding Dim | 256 |
| Proprioception | yes |

**Table 12:** Hyperparameters for Diffusion Policy, shared for all simulation experiments.

| | |
|---|---|
| Training Steps | 500K |
| Batch Size | 64 |
| Optimizer | AdamW |
| Learning Rate | 1e-4 |
| Weight Decay | 1e-6 |
| Learning Rate Schedule | Cosine Decay |
| Linear Warmup Steps | 1000 |

**Table 13:** Training Hyperparameters, shared for all simulation experiments.

