# OpenReview forum: "So You Think You Can Scale Up Autonomous Robot Data Collection?"
_robot-learning.org/CoRL/2024/Conference — CoRL 2024_

### Official Review · Reviewer_wA2S · 2024-07-19
**Interesting paper**

**Originality:** 4
**Technical Quality:** 3
**Clarity Of Presentation:** 4
**Potential Impact:** 3
**Recommendation:** 3
**Confidence:** 2

**Review:**

Strengths:
1. Investigating what would be a good way to scale up robot data collection and propose a criticism is valuable.
2. The paper is well-written and easy to understand.

Weaknesses:
1. In Table 2, it would make this paper stronger if the authors could show a plot that shows the performance trend given different amounts of near-failure H data and random H data (maybe also do the same thing for autonomous data). Through this plot, we could get the following insights:

    a) How much data does near-failure H data need to get the same performance as simply adding random human data?

    b) How will the performance change if we keep adding near-failure human data?

    c) How much data do we need to increase a fixed percent of a performance increase (such as 10%)? For example, from 50% to 60%, 60% to 70%, 70% to 80%, how much data do we need separately? At which point, increasing a fixed percent of performance would need an incredible amount of human data or would need data that we can not afford. At that point, it might be the right time to add autonomous data. To get better performance, we know that adding human-annotated data is a simple but useful way, but we are not sure when we should stop and switch to autonomous data.

2. In the supplementary section, I found that for each task, the authors collected at most 100 human data; some of them only had 5 human data, and at most 500 autonomous data, some of them less than 100 autonomous data. I am not sure experiments based on such an amount of data could make convincing conclusions. The main idea I got from the experiments is that when the data amount is small, simply adding human data is useful, especially adding demonstrations that correct previously failed trajectories.

Discussions:
1. My hypothesis why autonomous data does not benefit the performance is because those successful trajectories are basically in-domain and only used to reinforce success trajectories, but it does not help with those failed trajectories. That is why human data helps, since compared with autonomously collected data, human data is more likely to be out of the domain and bring something useful for training.

2. For scaling up problems, I think we should always think of building based on previous work and building a model that people can keep contributing to. This idea is similar to people working on large language models. For me, the key to making autonomous IL successful is designing a decent and able-to-continue improving success detectors and a framework or model that could autonomously correct failed trajectories.

**Quality Of The Limitations Section:**

3

**Questions For Rebuttal:**

Check the weaknesses section above.

**Robotics Focus:**

4

**Summary Of Paper:**

This paper investigates how to scale up robot data collection and mainly focus on autonomous imitation learning (IL).

**Summary Of Recommendation:**

This idea of this paper is interesting, but I think deeper investigation is still needed.

---

### Official Review · Reviewer_BvKu · 2024-07-20
**Useful, insightful, but limited analysis and missing novelty**

**Originality:** 1
**Technical Quality:** 3
**Clarity Of Presentation:** 5
**Potential Impact:** 2
**Recommendation:** 3
**Confidence:** 3

**Review:**

## Strengths
* **Presentation**: the work reads well and the flow is good. I think the justification for this study is provided clearly and that the results and analysis are presented clearly
* **Insightful**: I think the study provided is overall useful. Despite the insights coming only from empirical evaluation, the analysis helps to understand some of the fundamental challenges that come from RL, IL or the recently proposed autonomous IL paradigms, when applied in robotics (manipulation!)
## Weaknesses
* **Novelty**: there is no novelty introduced in this work, which mainly represents a benchmark of previous approaches and of the authors' ideas to solve the challenges presented. While this is not necessarily motivation for rejection, it should be accounted
* **Limitedness**: as also stated in the Limitations section by the authors, the study focuses on a specific configuration of the agent (fixed architecture and policy class). I am particularly concerned with the policy class being fixed (diffusion policy) as this has a major impact on the execution, and thus on future data collection for autonomous IL

**Quality Of The Limitations Section:**

3

**Questions For Rebuttal:**

* I am not sure about the usefulness of Figure 1. While it is used to support the Introduction of different paradigms, it is somehow hard to interpret at first (before reading the full Section) and it is not really scientific (the curves are just "drawn")
* In order to be more rigorous, I think the work should have analyzed better the causes of the data collected by autonomous IL being unable to improve performance. Some examples of this could have been: (i) analyzing different imitation learning strategies (e.g. ACT [1] would have been a nice addition), (ii) analyzing the data distribution and comparing it with the human demonstrations, rather than focusing solely on data scaling.

[1] Learning Fine-Grained Bimanual Manipulation with Low-Cost Hardware, Zhao et al

**Robotics Focus:**

4

**Summary Of Paper:**

The work presents an analysis of the autonomous imitation learning (IL) paradigm, mainly highlighting the limitations of this approach. In particular, the work shows that autonomous IL suffers from scaling issues similar to those of other IL and reinforcement learning (RL) paradigms, while offering marginal performance improvements.

**Summary Of Recommendation:**

I think the work could be useful, but at the moment I think the study is too limited for being convincing about the insights provided.

---

### Official Review · Reviewer_7gqs · 2024-07-20
**This paper tackles a very important and relevant problem, that of scaling up robot learning. By improving the writing and clarity of the paper I believe this work can be  impactful.**

**Originality:** 3
**Technical Quality:** 5
**Clarity Of Presentation:** 3
**Potential Impact:** 3
**Recommendation:** 3
**Confidence:** 4

**Review:**

Strengths

This paper addresses a very important problem that is very relevant to the current work in the robot learning community: that is, how difficult it is to scale up robot learning and also the importance of different data, e.g., human vs auto data in policy performance. The authors have done a great job analysing a large number factors that affect autonomous IL covering both environment design, e.g., reset functions, success detection and human supervision, e.g., different sizes of human supervision, active learning, etc. Additionally, the tasks chosen cover a range of complexity in manipulation and the number of evaluations performed is impressive which I think is very important in such large scale study. Overall, the insights drawn from the paper are going to be of interest to the general robot learning community and I enjoyed reading about them.

Weaknesses

The main weakness of the paper is that is writing becomes confusing. It is often hard to understand exactly what method is used to answer each question addressed at the different sections of the paper. For example, for the hanging mitts task (sec 3.3),  how many demonstrations were used for diffusion policy for that task (I can see that for the sec. 4 tasks this is provided in the appendix). For the reset study in sec. 3.4 how was the reset policy actually learned? And for the image classifier of sec 3.2, what did you implement exactly to obtain the results stated? I think understanding how each of these components was implemented in detail would be very important for a study paper like this. Also, you mention that success detection for the mitts task was straightforward but it is unclear what you actually used to autonomously detect successes in the end, was it a classifier?

In a similar manner I think that section 4, needs to be structured better to be more clear. For example, what was the success detection used for the tasks in section 4.1. Every how many roll-outs were successes detected and autonomous data collected? I think this would give a good estimate of the time that would be needed for an autonomous IL method across different tasks. How did you make sure that no false positives were actually included in the autonomous data in the end for the sec. 4 experiments? And if false positives were included in the data, how did they affect learning? This is a problem nicely brought up in sec. 3, but it is unclear how it is addressed in sec. 4. Finally, I think videos on the website showing how each of the different studies was conducted would be very helpful here.

**Quality Of The Limitations Section:**

3

**Questions For Rebuttal:**

Please see the questions asked above.

**Robotics Focus:**

4

**Summary Of Paper:**

This paper studies autonomous IL as a way to strike a good balance between autonomous robot learning and reduced human effort to reset environments, design reward functions or the need to provide multiple human demonstrations. The paper studies a very important and relevant problem to the current robotics community that provides insights on the difficulties of scaling up robot learning efficiently with respect to human effort. The paper provides  very detailed and well evaluated experiments on a variety of manipulation tasks that range in complexity. The authors study a wide range of different factors that affect scaling up autonomous IL that involve environment design: designing success detectors, reset mechanisms etc… and human supervision factors: e.g., studies on different scales of human and autonomous data, different training techniques, different data collection methods, such as active learning, the importance of using failure data for offline RL etc. The authors conclude that providing several more human demonstrations, especially from failed initial states is often more effective than any of the other data collection methods which often require considerable human effort as well.

**Summary Of Recommendation:**

I opted for WR mainly because I think the paper requires some work to improve its clarity. If this is addressed I am happy to raise my score as the paper's scope is important.

---

### Author Rebuttal · Authors · 2024-08-11

We have attached a revised version of our submission (changes in blue text) incorporating the reviewers' feedback. The main paper and appendix are in one file. Please see the Overall Response for a summary of the changes, and comments on the individual reviews for point-by-point responses to the reviewers' comments. Thank you all!

---

### Decision · Program_Chairs · 2024-09-04

**Decision:**

Accept

**Comment:**

The paper presents a study of autonomous IL methods across different scales, simulation environments and real-world tasks and contrast this approach with human data collection.

Strengths:
- Well-written paper addressing a relevant problem in data collection for robot learning
- Interesting insights from a broad empirical evaluation

Weaknesses:
- Methodology requires improvement. Sometimes it is difficult to infer what method and parametrization are being analyzed
- More details needed in the experiments to fully draw the conclusions
- Potentially not enough human data used in the experiments

The reviewers provided valuable suggestions to improve the paper. I encourage the authors to work with the reviewers in addressing their concerns.

=======================================
Pos-rebuttal update:

Thank you for addressing the reviewers' comments in the revised version. They now agree that the paper should be accepted.